# Classification of Precipitation Types Based on Machine Learning Using Dual-Polarization Radar Measurements and Thermodynamic Fields

Kyuhee Shin [1,2], Kwonil Kim [1,2,*], Joon Jin Song [3] and GyuWon Lee [1,2]

1   Department of Atmospheric Sciences, Kyungpook National University, Daegu 41566, Korea
2   Center for Atmospheric REmote Sensing (CARE), Kyungpook National University, Daegu 41566, Korea
3   Department of Statistical Science, Baylor University, Waco, TX 76798, USA
*   Correspondence: kwonil@knu.ac.kr

**Abstract:** An accurate classification of the precipitation type is important for forecasters, particularly in the winter season. We explored the capability of three supervised machine learning (ML) methods (decision tree, random forest, and support vector machine) to determine ground precipitation types (no precipitation, rain, mixed, and snow) for winter precipitation. We provided information on the particle characteristics within a radar sampling volume and the environmental condition to the ML model with the simultaneous use of polarimetric radar variables and thermodynamic variables. The ML algorithms were optimized using predictor selection and hyperparameter tuning in order to maximize the computational efficiency and accuracy. The random forest (RF) had the highest skill scores in all precipitation types and outperformed the operational scheme. The spatial distribution of the precipitation type from the RF model showed a good agreement with the surface observation. As a result, RF is recommended for the real-time precipitation type classification due to its easy implementation, computational efficiency, and satisfactory accuracy. In addition to the validation, this study confirmed the strong dependence of precipitation type on wet-bulb temperature and a 1000–850 hPa layer thickness. The results also suggested that the base heights of the radar echo are useful in discriminating non-precipitating area.

**Keywords:** precipitation type; machine learning; dual-polarization radar; thermodynamic field

## 1. Introduction

Precipitation types are a major factor in weather forecasting as they affect transportation, agriculture, and industry as a whole [1,2]. Accurate discrimination among winter precipitation types (e.g., rain, snow, wet snow, and freezing rain) plays a key role in winter precipitation forecasting. An unpredicted phase transition from rain to snow, caused by small changes in the thermodynamic environment, can result in disruptions in the road and air traffic and create hazardous walking conditions. The accurate determination of surface precipitation type is also important to reduce uncertainty in hydrological models. However, the significant sensitivity of precipitation type to the variability in the atmospheric condition is one of the major challenges in the accurate classification of the type at the surface.

Several meteorological variables control the winter precipitation type. Air temperature ($T_s$) is the most fundamental variable for classifying precipitation types [3,4]. In addition to the $T_s$, wet-bulb temperature ($T_w$) is known as a more critical variable than $T_s$ [5–8] because it accounts for the cooling effects by surface evaporation [9]. Thus, $T_w$ generally improves the classification of the precipitation type. Relative humidity (RH) measures the amount of moisture and affects particle size distribution [5,10,11]. The thickness of the 1000–850 hPa layer ($Thick_{1000-850}$) is proportional to the mean temperature of the layer. The thickness was also considered as a key variable in the discrimination between rain and snow [12–15].

The low-level vertical temperature lapse rate ($\gamma_{low}$) explains the conditions in which the particles fall [6].

Based on these fundamental variables, the critical RHs as a function of $T_s$ for the judgment of the precipitation type were obtained by statistical analysis [16] (so-called Matsuo scheme). The optimized Matsuo scheme [15], which is suitable for the Korean climate, was utilized in the Korea Meteorological Administration (KMA) operationally to support the decision making of the forecasters. In practice, the scheme is applied using the variables of the numerical weather prediction (NWP) models. Thus, the accuracy of the classification of precipitation types strongly depends on that of the NWP [17]. A low spatiotemporal resolution also causes the low predictability of the scheme.

Aside from the thermodynamical parameters, the advent of a dual-polarization radar network offers an opportunity to utilize dual-polarization variables for accurate precipitation type classification at a high spatiotemporal resolution. The dual-polarization variables include differential reflectivity ($Z_{DR}$, dB), specific differential phase ($K_{DP}$, $° \ km^{-1}$), and the co-polar correlation coefficient ($\rho_{HV}$). In the perspective of precipitation classification, $Z_{DR}$ and $K_{DP}$ are useful to determine the particle shape, while $\rho_{HV}$ is effective in determining the melting layer [18,19]. A hydrometeor classification algorithm using polarimetric variables was applied using a fuzzy-logic approach [20–22], a Bayesian approach [23,24], and so on. In particular, the fuzzy-logic algorithms were utilized for operational use and their classifications over the past 20 years are displayed in realtime in several national radar networks [25–28] to the present. Although the dual-polarization variables can provide detailed hydrometeor classification at the high spatiotemporal resolution, the absence of information (particularly thermodynamical information) below the radar beam height leads to large uncertainty in determining the precipitation type at the surface.

The synergistic use of polarimetric variables and thermodynamic variables is beneficial in the accurate classification. Schuur et al. [29] first introduced an algorithm that combines S-band polarimetric radar observations and $T_w$ obtained from the NWP. Based on the defined Boolean decision tree, the algorithm classifies the winter precipitation phases into seven classes (crystals, dry snow, wet snow, ice pellets/sleet, freezing rain, a mix of freezing rain and ice pellets, and rain). They generated background classification from NWP and updated it using radar data. The validation with the data from an automated surface observing system (ASOS) showed an encouraging performance. Recently, Steinert et al. [30] determined the hydrometeor type at the radar beam height using a fuzzy-logic hydrometeor classification algorithm and took into account the melting process below the beam height using the NWP product based on a Boolean decision tree. Despite these efforts, the decision tree is still less objective and less flexible due to empirical critical thresholds (e.g., reflectivity, $T_w$, melting layer height).

Machine learning (ML) perfectly fits this particular need for modeling the complex relationship between the precipitation type and the various factors. It allows us to deal with multidimensional and complex nonlinear problems without any distributional and modeling assumptions. For these benefits, ML was widely used in the atmospheric sciences [31–33]. Using the NWP-driven variables, a multinomial logistic regression (MLR) model showed a 15% improvement in accuracy compared to the NWP forecast and outperformed the optimized Matsuo scheme [34]. Seo [35] utilized both thermodynamic variables and polarimetric variables and performed testing with the following six ML models: k-nearest neighbors, logistic regression, support vector machine (SVM), decision tree (DT), random forest (RF), and multi-layer perceptron. As a result, RF showed the best score and performed better than the operational method. Półrolniczak et al. [36] implemented the RF using the surface synoptic observation, NWP reanalysis data, and radar data in the similar way. However, there are opportunities to improve the performance through the optimization of each ML model. To efficiently implement the ML model in realtime operation, it is necessary to optimize and search for a suitable ML model through rigorous evaluation to ensure both accuracy and computational efficiency in realtime operation.

In this paper, we explored the capability of the optimized ML to classify the precipitation types using both polarimetric and thermodynamic variables. Three widely used supervised ML methods (DT, RF, and SVM) were tested. The optimization of the ML algorithms was performed by (1) selecting predictors through variable importance analysis and (2) tuning each ML model with the best hyperparameters. The ML models were evaluated using the hold-out validation. The best ML model was compared with the schemes in operation and applied to the Korean S-band operational radar network.

## 2. Response and Predictors

To establish the ML model, it is necessary to properly select the response variables and predictors. In our configuration, the response variable is the precipitation type, and the predictors are both polarimetric variables and thermodynamic variables.

### 2.1. Precipitation Type

Trained observers at KMA observed the present weather, which was recorded and assigned to one of the weather codes. The weather codes were reported at 23 stations (blue circles in Figure 1). While automatic instruments (i.e., disdrometers and present weather sensors) provided weather codes and could be used for training and validation [37], this record was deemed the most reliable source of precipitation in Korea as it was collected by a trained observer. Each observer should follow the guidelines in the observation manual of KMA to ensure the consistency of the quality from site to site.

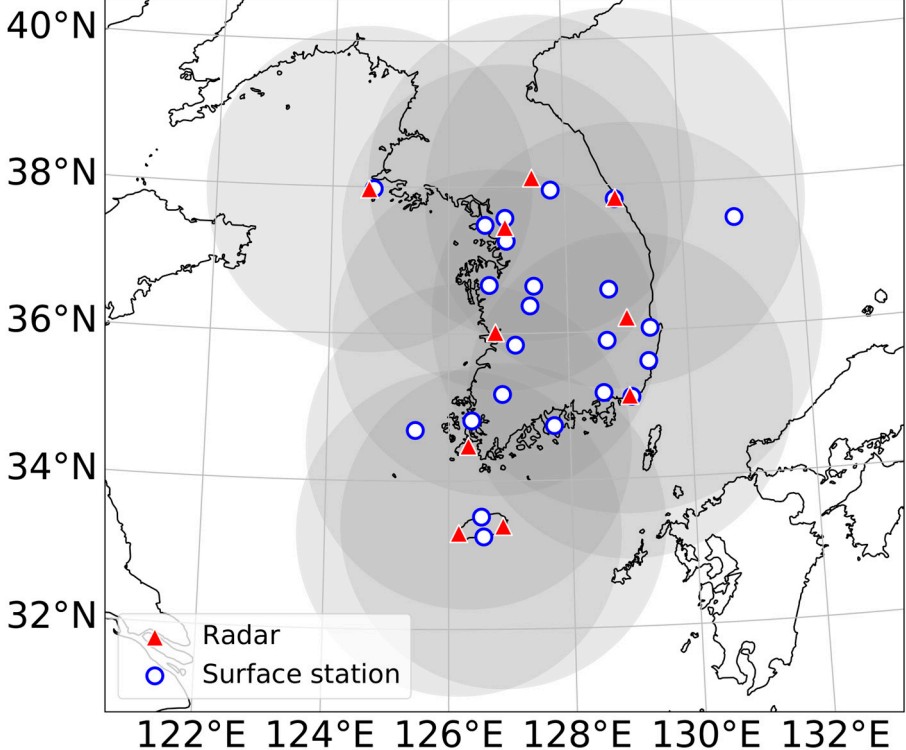

**Figure 1.** Dual-polarization radar networks (red triangle) with ~240 km range from each radar (gray shading) and surface station for present weather observation (blue open circle).

The weather code consists of hydrometeor, lithometeor, photometeor, and electrometeor phenomena with 143 codes [38]. Among the codes, we selected 3 codes for rain (RA; rain, drizzle, and rain shower), 3 codes for snow (SN; snow, snow shower, and graupel), and 2 codes for mixed type (MIX; sleet and sleet shower). The mixed type here was defined as the co-presence of liquid (rain) and solid (snow) phases. The hail was excluded because it had never been reported in the training dataset and thus training was not possible. The

other codes were considered as no precipitation (CLR), indicating the absence of precipitating particles at the surface. As a result, we divided the present weather codes reported by trained people into the following four types: rain (RA), snow (SN), mixed (MIX), and no precipitation (CLR).

### 2.2. Dual-Polarization Radar Data

The polarimetric radar variables were obtained from the radar network of 10 S-band radars [39,40] operated by the weather radar center (WRC) of the KMA. The variables include $Z_H$, $Z_{DR}$, $K_{DP}$, and $\rho_{HV}$. The quality of radar variables was controlled by a KMA algorithm, named the clutter elimination algorithm for the non-precipitation echo of radar data (CLEANER) [41]. The CLEANER algorithm features the efficient removal of a non-precipitation echo and allowed us to use reliable data during training.

A national 3D radar mosaic product of each radar variable has a horizontal resolution of 500 m and a vertical resolution of 50 m. To use the collocated variables as input data for ML, if any of the radar variables is missing, we considered the corresponding grid point as missing. Thus, we used data from grid points where all variables were valid. Then, we took the value of each variable at the lowest height where meteorological echo appeared for each column (i.e., the height of echo base) to reflect the characteristics of radar variables closest to the surface. We also included the height of the echo base as the predictor referred to as HGT.

Note that one of the radars (located in the most northeast in Figure 1) was replaced by a polarimetric radar from the Doppler radar in November 2019. Therefore, before November 2019, $Z_{DR}$, $K_{DP}$, and $\rho_{HV}$ were mosaicked by 9 radars excluding this radar, whereas $Z_H$ was mosaicked by 10 radars.

### 2.3. Thermodynamic Field Data

The three-dimensional thermodynamic fields, which are operationally produced by KMA every 5 min, were used to retrieve thermodynamic predictors. The horizontal resolution is 4 km, and the vertical resolution is 100 m at 0–2 km height and 200 m at 2–10 km height. The fields are based on the product from the very-short-range forecast system of KMA and updated by multiquadric interpolation of observations [39,42]. The available variables include dry-bulb temperature (T), dew point temperature ($T_d$), and pressure. The predictors were obtained from these three variables.

This study took into account the measurements at the surface and the vertical profile of the variables. In regard to the variables at the surface, we computed $T_s$ and $T_w$. $T_s$ (temperature nearest to the surface) was obtained from the T at the lowest level. $T_w$ was calculated using $T_s$ and RH via Stull [43], where RH was calculated from $T_s$ and $T_d$ at the lowest height using the "MetPy" package [44] in Python. Variables derived from the atmospheric profiles include $Thick_{1000-850}$ and $\gamma_{low}$. $Thick_{1000-850}$ was computed using the hypsometric equation for the layer 1000–850 hPa. We used the virtual temperature at 925 hPa for computational purposes, although the one in the equation is the layer-mean virtual temperature. Thus, $Thick_{1000-850}$ (gpm) is expressed as

$$Thick_{1000-850} = \frac{R_d \overline{T_v}}{g} \ln \frac{P_1}{P_2} \tag{1}$$

where $R_d$ is the gas constant of dry air (287 J kg$^{-1}$ K$^{-1}$), g is the gravitational acceleration (9.8 m s$^{-2}$), and $P_1$ and $P_2$ are 1000 hPa and 850 hPa, respectively. $\overline{T_v}$ is the virtual temperature (K) at 925 hPa.

The $\gamma_{low}$ (K km$^{-1}$) was calculated from the difference of T in the lowest 500 m height [6] as

$$\gamma_{low} = -\frac{\Delta T}{\Delta z} = -\frac{T_{500m} - T_s}{0.5} \tag{2}$$

where $T_{500m}$ is the temperature at 500 m height.

Since the horizontal resolution of dual-polarization radar measurements and thermodynamic fields are not consistent, linear interpolation was applied to the three-dimensional analysis fields prior to the computation of predictors to make the grid size the same as the dual-polarization radar data. A total of 12,367 10 min data during the winter season 2018–2019 was used for training and validation. A total of 16 days was selected including 3 days for rain-dominant, 9 days for the co-occurrence of rain, mix, and snow, and 4 days for snow-dominant cases (Table 1). In addition to the dataset for training and validation, the 5 independent cases in 2022 were selected for the application in the operational radar data.

**Table 1.** List of precipitation events for training, validation, and application.

| No. | Date | Configuration | No. | Date | Configuration |
|---|---|---|---|---|---|
| 1 | 13 December 2018 | | 12 | 15 February 2019 | |
| 2 | 16 December 2018 | | 13 | 16 February 2019 | Training |
| 3 | 23 December 2018 | | 14 | 18 February 2019 | and |
| 4 | 27 December 2018 | | 15 | 19 February 2019 | Validation |
| 5 | 28 December 2018 | Training | 16 | 27 February 2019 | |
| 6 | 29 December 2018 | and | 17 | 25 January 2022 | |
| 7 | 12 January 2019 | Validation | 18 | 15 February 2022 | |
| 8 | 19 January 2019 | | 19 | 19 February 2022 | Application |
| 9 | 31 January 2019 | | 20 | 1 March 2022 | |
| 10 | 3 February 2019 | | 21 | 19 March 2022 | |
| 11 | 7 February 2019 | | - | - | - |

## 3. Classification Methods of Precipitation Types

Figure 2 illustrates the overall procedures of the construction of our final ML model.

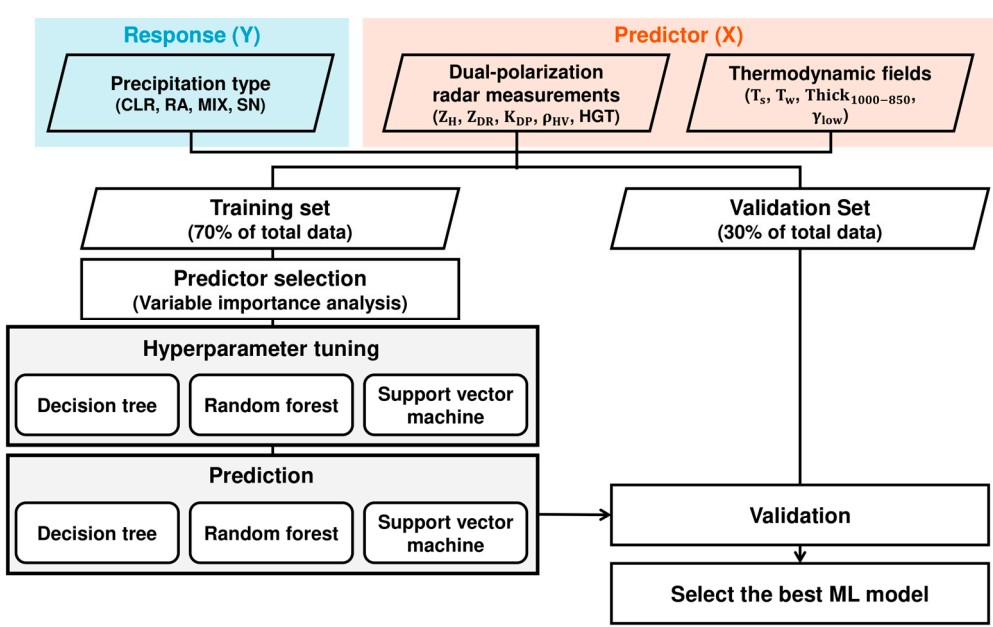

**Figure 2.** A flow chart for constructing the proposed machine learning model.

### 3.1. Machine Learning Methods

This study examines the effectiveness of supervised-ML methods in classifying the winter precipitation types. We considered the following three widely used classifiers: DT, RF, and SVM. The DT method is a building block of all tree-based methods. It summarizes splitting conditions used to segment the predictor region in a tree shape [45]. The DT algorithm starts by dividing the tree recursively from the top of the tree as the root node. The best predictor for splitting was selected at each node and the data were divided into

the sub-node called the intermediate node. To perform this splitting, the algorithm applies an empirical rule to decide the criteria based on the lowest impurity measure, such as the Gini index, the information gain, or the gain ratio. In this study, the Gini index was chosen for splitting nodes. The node where there is no more data to split is known as the terminal node.

RF is an ensemble model and more robust to noise in training data [46]. It consists of a number of DTs built with bootstrap samples which were generated by sampling with replacement. Each DT was grown individually using the randomly selected subset of predictors. It can reduce the variance of RF by decorrelating DTs. The final prediction was obtained by pooling predictions from all DTs. DT and RF are known as a "white box" model, which enables the interpretation of the model [47].

The SVM generally finds a hyperplane that maximizes separating the data points to their potential classes [48]. The hyperplane is defined as the boundary that divides the classes of input data. The points that are closest to the hyperplane and used to calculate the margin are called support vectors. The technique measures the distances between a new observation and each support vector with a selected kernel function and makes its prediction. In this study, we explored several kernel functions (linear, polynomial, and radial basis function) and then selected the polynomial functions for classification.

### 3.2. Predictor Selection

In this study, a total of nine variables were regarded as the candidate predictors, which include both polarimetric ($Z_H$, $Z_{DR}$, $K_{DP}$, $\rho_{HV}$, and HGT) and thermodynamic ($T_w$, $Thick_{1000-850}$, $T_s$, and $\gamma_{low}$) variables. Through a variable importance analysis, we examined the effect of each predictor to classify the precipitation types and selected the final predictors. The variable importance was measured by mean decrease Gini (MDG) values and was computed using the "randomForest" package [49] in R software. The MDG values are the sum of the total decrease in the Gini index from splitting on a given feature for a tree and are averaged over all trees. A large MDG value indicates an important predictor for classification.

The analysis indicated that $T_w$ is the most important predictor for classifying the winter precipitation types (Figure 3). This is consistent with previous works that highlighted that $T_w$ is a more significant factor in distinguishing the precipitation types than dry-bulb temperature [5,7,8], as shown by a higher MDG value (1503.05) compared to $T_s$ (851.16). It is interesting to note that HGT is more important than the other polarimetric variables. $\rho_{HV}$, which is useful in hydrometeor classification, ranked third among the polarimetric variables. However, $Z_{DR}$ and $K_{DP}$ have the lowest importance among the candidate predictors, with MDG of 318.27 and 232.80, respectively. Thus, we excluded $Z_{DR}$ and $K_{DP}$ from the predictors. In other words, the final set of the predictors for the ML models includes three polarimetric ($Z_H$, $\rho_{HV}$, and HGT) and four thermodynamic ($T_w$, $Thick_{1000-850}$, $T_s$, and $\gamma_{low}$) variables.

### 3.3. Hyperparameter Tuning

There are several tuning parameters for each ML model. We optimized the ML models by choosing a set of optimal hyperparameters. The ML models were trained by using the selected predictors in the training and validation dataset.

DT can be optimized by reducing the depth of trees through pruning. The pruning reduces the complexity of the DT, improving the prediction accuracy by removing the overfitted subtrees. We conducted pruning using the argument called complexity parameter (cp) in the "rpart" package [50] in R software. It is based on the cost-complexity algorithm to find the optimal depth of the tree [45]. The optimal size of trees is typically chosen when cp is near 0.01. If the value is smaller than 0.01, DT tends to be complicated and overfit. The size of the tree (and corresponding cp value) is displayed in Figure 4 with the 10-fold cross-validation error (i.e., X-val relative error). The DT was appropriately pruned by choosing seven leaves to be the size of the tree.

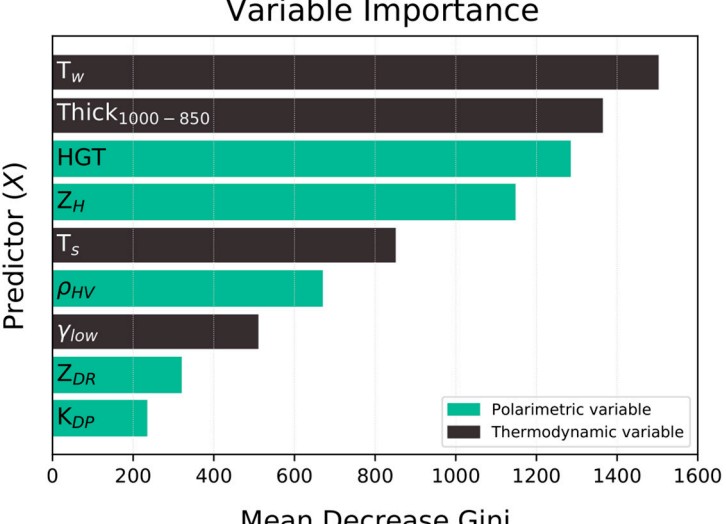

**Figure 3.** The importance of the predictors (mean decrease Gini) with importance decreasing from top to bottom. Polarimetric variables are in green and thermodynamic variables are in gray.

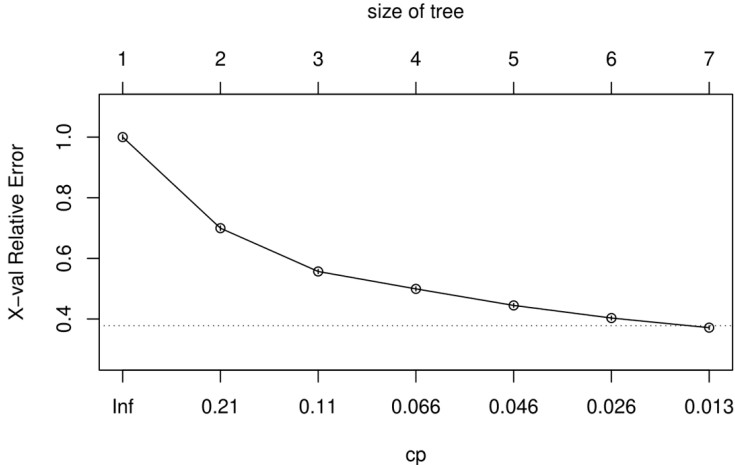

**Figure 4.** X-val relative error of the DT model as a function of the different sizes of tree and corresponding cp value.

RF has two hyperparameters: the number of trees ($n_{tree}$) and the number of predictors ($m_{try}$). Predictors are randomly sampled in each DT. Liaw and Wiener [49] suggested $n_{tree}$ of 500 and $m_{try}$ of $\sqrt{p}$ as the default value for a classification problem, where p is the total number of predictors. The default value of $m_{try}$ is 3 ($\approx \sqrt{7}$) since p is 7 in this study. To determine the optimal values of $m_{try}$ and $n_{tree}$, we calculated the out-of-bag error (OOB error) by varying the $m_{try}$ from 1 to 7 and $n_{tree}$ from 100 to 1500 (Figure 5). The local minimum of OOB error (0.092) was found at $m_{try}$ of 4 and $n_{tree}$ of 500. The OOB error decreased from about 0.098 to 0.092 until $m_{try}$ was 4 when $n_{tree}$ was 500 then tended to increase with increasing $m_{try}$. The global minimum appeared at $n_{tree}$ of 1500 and $m_{try}$ of 3, but time efficiency must be considered given the similar OOB error but longer computational time compared to $n_{tree}$ of 500. Therefore, we selected $m_{try}$ of 4 and $n_{tree}$ of 500 as optimal values.

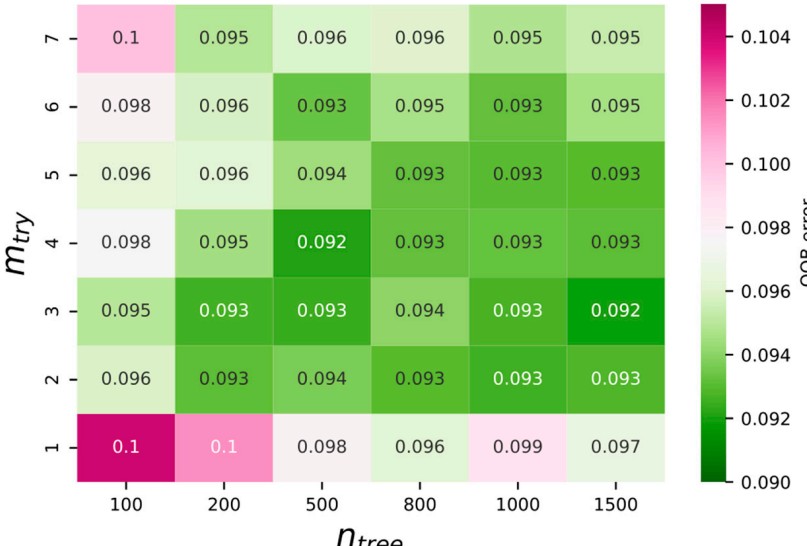

**Figure 5.** The OOB error as a function of the number of predictors ($m_{try}$) and the number of trees ($n_{tree}$).

Lastly, SVM can be optimized using the cost and gamma values where the best model performance is achieved. The cost specifies the width of the margin. The cost results in a trade-off between high accuracy and smoothing decision boundary. A larger cost leads to a narrower margin (i.e., lower error) but might also increase the chance of overfitting. The gamma determines the distance of influence of a single training point. High gamma reduces the distance of influence as closer points have more weight. We evaluated the SVM model with various combinations of cost and gamma and examined the 10-fold cross-validation error rate (Figure 6). We found the optimal cost value of 0.1 and the gamma value of 3, where the lowest error rate (0.14) occurred.

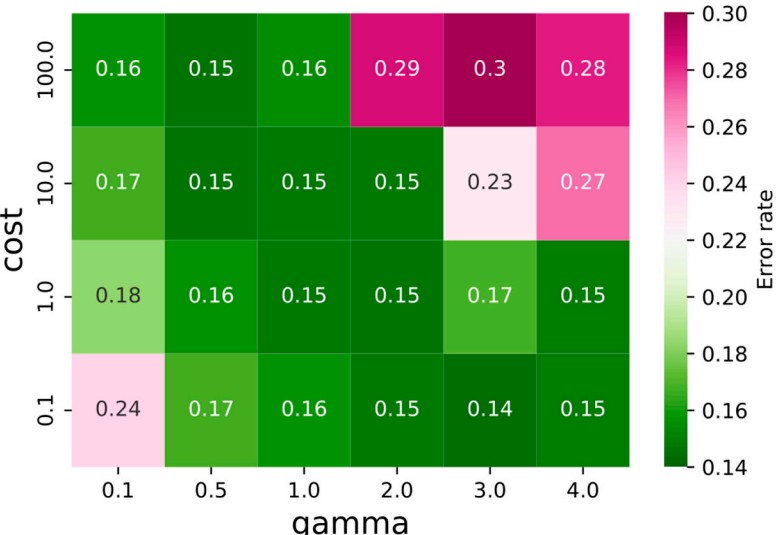

**Figure 6.** The 10-fold cross-validation error rate of the SVM model under the different cost and gamma. The value in each tile indicates the error rate.

### 3.4. Validation

The 70–30 hold-out validation was performed to compare the accuracy of ML models. The numbers of the training and validation data for each class are summarized in Table 2. A total of 8692 and 3675 10 min data were used for training and validation, respectively. It is also necessary to examine whether the ML model can provide a better classification

compared to the scheme in operation. Therefore, in addition to the hold-out validation, we compared the best ML model with the optimized Matsuo scheme (see Appendix A for details), which is currently used as a fundamental reference for forecasters to make decisions [15].

**Table 2.** The number of 10 min data for the training and validation.

|  | No Precipitation (CLR) | Rain (RA) | Mixed (MIX) | Snow (SN) | Overall |
|---|---|---|---|---|---|
| Training set | 3376 | 3721 | 128 | 1467 | 8692 |
| Validation set | 1425 | 1572 | 45 | 633 | 3675 |
| Total | 4801 | 5293 | 173 | 2100 | 12,367 |

To assess model accuracy, we used the probability of detection (POD), false alarm rate (FAR), and critical success index (CSI). The POD, FAR, and CSI were derived based on $4 \times 4$ contingency metrics. The perfect score of POD and CSI is 1, whereas the perfect score of FAR is 0. They are defined as follows:

$$POD = \frac{H}{H + M} \tag{3}$$

$$FAR = \frac{F}{H + F} \tag{4}$$

$$CSI = \frac{H}{H + M + F} \tag{5}$$

where a hit (H) occurs when the predicted type agrees with the observation, a miss (M) happens when a precipitation type was observed and the model predicts another type, and a false alarm (F) represents the case when the predicted type does not occur.

## 4. Verification of Precipitation Type Classification

We evaluated the performance of ML models for each precipitation type using 70–30 hold-out validation. All ML models showed a high POD for CLR, RA, and SN (Figure 7). Among the types, RA showed the best POD (0.931 for RF, 0.926 for SVM, and 0.882 for DT). This is not surprising given the huge amount of training data of RA. Despite the smaller amount of training data (less than half of RA), SN presented comparable performance to RA. In the case of MIX, all ML showed low scores (e.g., CSI less than 0.3). The smaller number of training data (1.47% of total training data) could be one of the primary reasons, but strong variability of the atmospheric condition in time and space with MIX [17] also makes the classification difficult.

For all precipitation types, the RF had the best score, followed by the SVM and the DT. The RF presented a POD of higher than 0.9 and a CSI of higher than 0.8 for both RA and SN. The RF also showed a better skill in the classification of MIX, as shown by the CSI of ~0.3, which is definitely higher than that of SVM (0.120).

As mentioned above, it is important to investigate whether ML models can perform better than the scheme in operation. The best ML model (RF) was compared with the optimized Matsuo scheme. Since the optimized Matsuo scheme (MS) does not consider no precipitation, CLR was excluded from validation.

As shown in Figure 8, the RF outperformed the optimized Matsuo scheme in the overall statistics. Both methods presented good performance in RA and SN. In particular, the PODs of the RF for RA and SN were almost equal to 1. In the MIX case, the RF showed a POD, FAR, and CSI of 0.441, 0.250, and 0.385, respectively. Although MIX is the most challenging type to predict, the RF reduced FAR by 0.742 (from 0.922 to 0.250) and increased CSI by 0.316 (from 0.069 to 0.385) compared to the optimized Matsuo scheme.

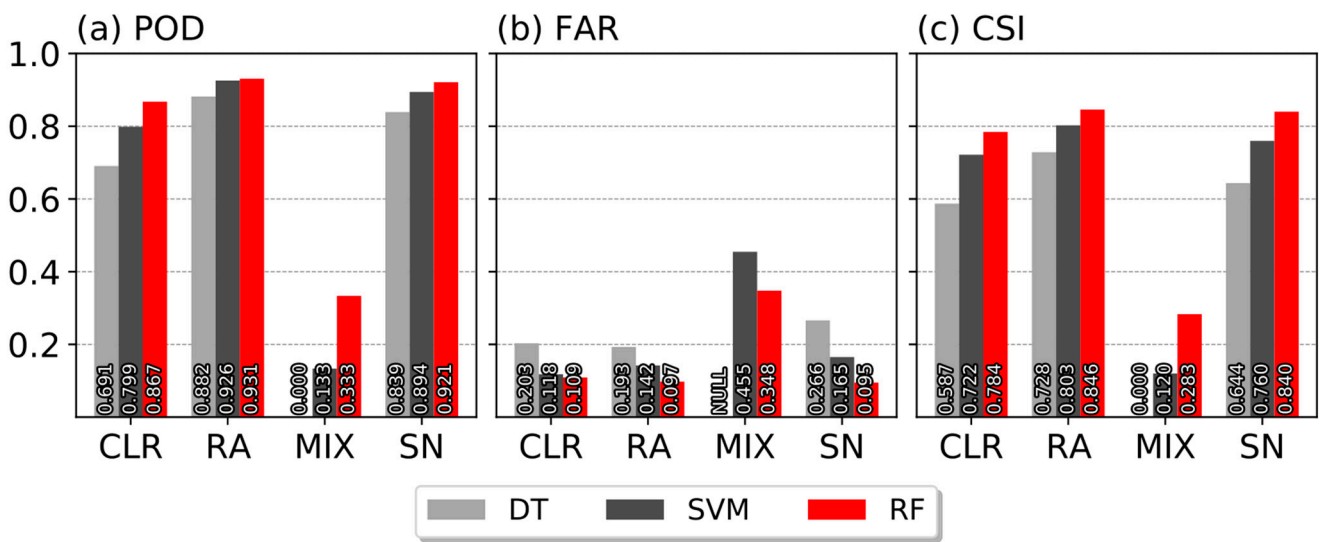

**Figure 7.** Skill scores of ML models, including (**a**) POD, (**b**) FAR, and (**c**) CSI.

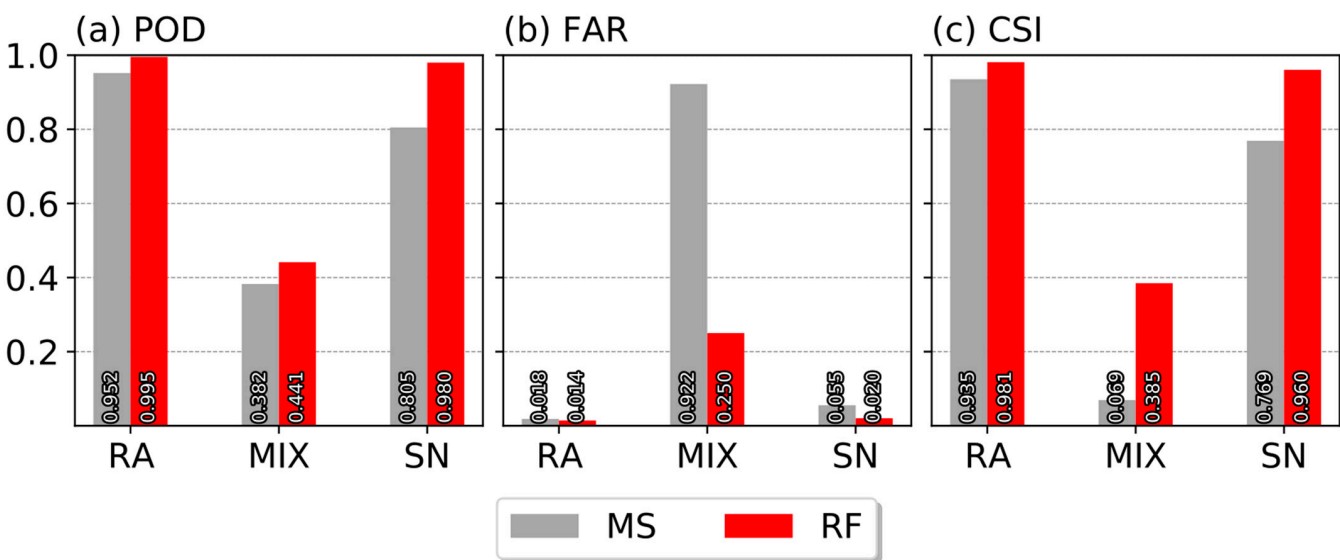

**Figure 8.** Same as Figure 7, but for the optimized Matsuo scheme (MS) and RF with the precipitation events.

## 5. Application to the Operational Radar Network

We applied the best model, RF, to the Korean S-band operational radar network. The aim of this part was to present some examples of how the selected ML model can be applied to the operational radar network. The classification of precipitation types by RF over the domain is shown in Figures 9 and 10. Note that the KMA expanded the domain of the national 3D radar mosaic product and included four more S-band radars in 2022.

In the rain-dominant case at 0310 LST, 1 March 2022, a widespread precipitation system was classified into either rain or no precipitation (Figure 9a). The areas where RF predicted rain had good agreement with the ground truth. It was also noticed that the regions of no precipitation (see the zoom-in-box) are consistent with the surface observation. A good agreement with the ground truth indicated the good capability of the proposed model to identify the region of no precipitation (in addition to rain) in a widespread precipitation system.

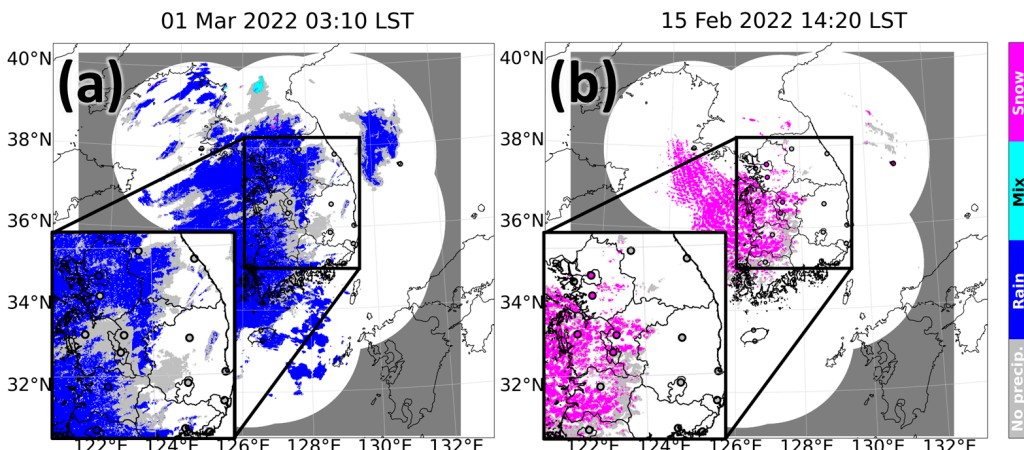

**Figure 9.** Examples of RF classification in (**a**) rain-dominant case (0310 LST, 1 March 2022) and (**b**) snow-dominant (1420 LST, 15 February 2022). The circle filled with color indicates the ground truth. The insets in (**a**,**b**) are zoom-in-boxes, providing a clear view of the classification.

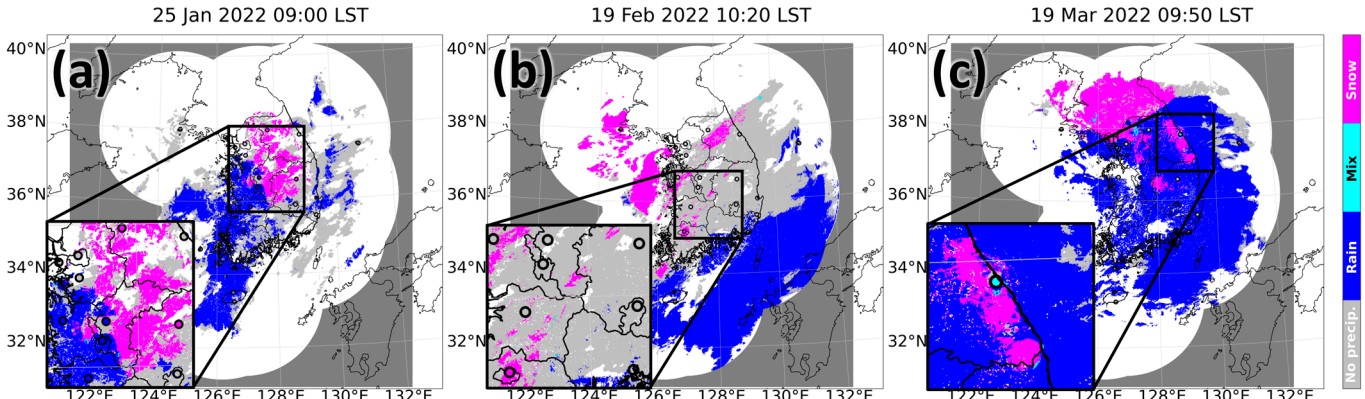

**Figure 10.** Same as Figure 9 except for cases of the co-occurrence of multiple precipitation types at (**a**) 0900 LST, 25 January 2022, (**b**) 1020 LST, 19 February 2022, and (**c**) 0950 LST, 19 March 2022.

The spatial distribution of the RF classification for the snow-dominant case (1420 LST, 15 February 2022) is shown in Figure 9b. Widespread snowfall dominated across the western part of the Korean Peninsula. The RF model performed well to capture the transition from no precipitation (due to sublimation) to snow. This result implies that the RF will be helpful for forecasters to determine areas of no snow but weak or moderate reflectivity.

Several cases when multiple precipitation types occurred were also examined (Figure 10). While the classification was performed at high resolution, the transition of precipitation type (from rain to snow) was consistent with the observed type (Figure 10a) even in the region where different types of precipitation were distributed in a small area.

Another example shown in Figure 10b illustrates the performance of RF in a case when no precipitation dominated. Although moderate or weak reflectivity occurred in most areas of South Korea, the RF classified a large fraction of no precipitation (Figure 10b). The result is reasonable because it coincides with the no precipitation report from the surface station. The narrow region of snow (near the bottom left region of the zoom-in-box in Figure 10b) was detected by RF and agrees with the snow report. The last example (19 March 2022) showing the result in the area of the mountainous region (Figure 10c) represented better chances of snow in the NW-SE-oriented mountain range. Besides, mixed precipitation reported at the station located near the coastal region was accurately discriminated as MIX by the RF.

Note that the trained ML model is also computationally efficient. In our case, the classification for the whole Korean radar domain was completed in a few seconds on

Intel Xeon Gold 6154 CPU at 3.00 GHz with single core, which met the computational requirement for the real-time operation. This is an encouraging result given the difficulties in performing the satisfactory classification in realtime due to the trade-off between the complexity of the algorithm (generally better accuracy) and the computational efficiency. It should be noted that the proposed model requires a thermodynamical field (e.g., variables from the NWP model) as well as polarimetric radar variables. If one datum is not available, the model is not applicable.

## 6. Discussion

Our results indicated that $T_w$ and $Thick_{1000-850}$ were highly related to the precipitation types as selected in the top two variables from the variable importance analysis. This is in agreement with previous studies that recommended the use of $T_w$ for classifying the precipitation types [5–8] since it better reflected the actual temperature of precipitation particles. This can be also seen from the distribution of $T_w$ in the training dataset for each precipitation type (Figure 11a), which exhibited the distinctive dependence of RA, MIX, and SN on $T_w$. Except for CLR, the distribution of $T_w$ of RA, MIX, and SN showed almost no overlap with each other, indicating that $T_w$ acted as a critical factor. The narrow distribution of MIX (−1.95 to 2.64 °C) means $T_w$ can serve as a significant factor in differentiating the most challenging type (MIX) from others.

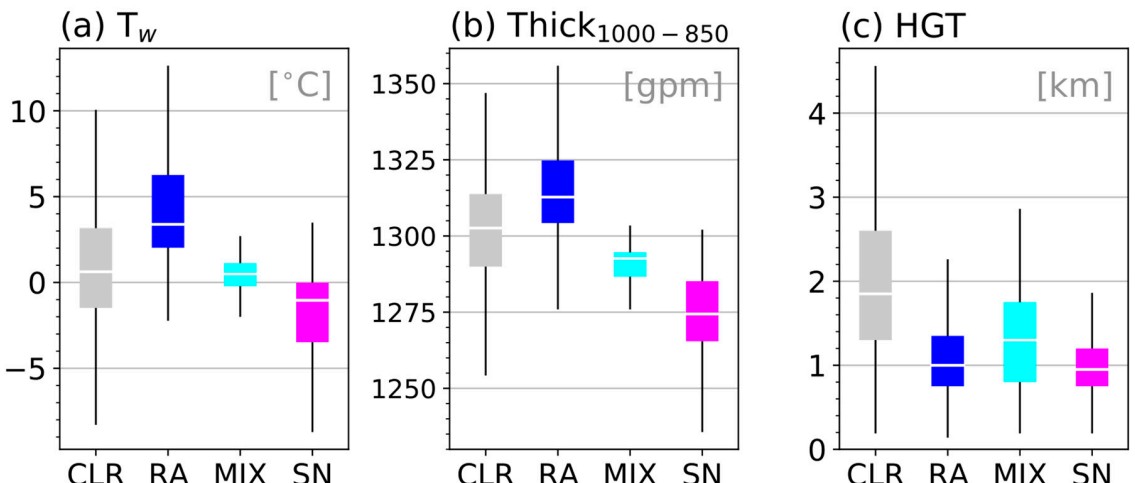

**Figure 11.** Boxplots of (**a**) Tw, (**b**) Thick$_{1000-850}$, and (**c**) HGT in terms of precipitation types using training data. A white horizontal line denotes the median, with color filled box extended from the first quartile to the third quartile.

Similarly, the distribution of $Thick_{1000-850}$, which reflects the mean temperature of the layer [12,15], showed strong dependence on the precipitation type (Figure 11b). Analogous to $T_w$, except for CLR, $Thick_{1000-850}$ was also distributed in different ranges of values according to the precipitation type so that they did not overlap with each other. This result indicates that both $T_w$ and $Thick_{1000-850}$ were the key factors for discriminating the precipitation types.

The presented ML model showed the reasonable identification of no precipitation areas. This could be helpful in many fields where accurate identification of the precipitation area is important such as transportation, agriculture, hydrology, and atmospheric science. To examine the evaporation or sublimation, one can utilize polarimetric variables [51–54] or/and the humidity profile from the NWP model [55]. Our study suggests that HGT can also be one of the useful factors in identifying the presence of precipitation on the ground (Figure 11c) without explicit representation of evaporation or sublimation. The HGT of CLR had higher values that are distinguishable from other types (Figure 11c). This is because the HGT implies the depth where evaporation or sublimation can occur "below the height of radar echo base".

There is still room for improvement in the model through careful treatment of noisy polarimetric variables, particularly $Z_{DR}$ and $K_{DP}$. Despite the lowest importance of $Z_{DR}$ and $K_{DP}$ in this study, the literature suggests that these variables are still believed to be good factors to differentiate between snowfall, melting layers, and rainfall regions [56–58]. The most likely reason for the low importance is the noisiness of $Z_{DR}$ and $K_{DP}$. With the reduction in the noisiness through a proper smoothing technique, the model was expected to have better performance by taking particle shape information into account.

Another thing that could lead to better performance of the model is the consideration of horizontal advection due to vertical displacement between the radar measurement and the surface, which can lead hydrometers to fall to another location on the ground. Although this study accounted for the radar variables at the lowest altitude, the location error by horizontal advection may be problematic in a strong low-level wind condition. The HGT may indirectly reflect the effect of the horizontal wind. However, one could obtain better predictions through an appropriate method to deal with this limitation.

In our study, we classified the precipitation type into CLR, RA, MIX, and SN. However, there are several important precipitation phases that we could not consider due to a lack of data such as freezing rain, ice pellets [58–62], and hail [63,64]. The ML model could be extended to include them when a sufficient number of data for these types are available.

## 7. Conclusions

Accurate classification of winter precipitation type at the surface is necessary for forecasters and researchers to minimize the damage of natural disasters on human life but is challenging due to its high variability in small changes in a thermodynamic environment. In this paper, to tackle this problem, we presented the ML-based winter precipitation type classification algorithm. The presented model attempts to exploit both polarimetric and thermodynamic variables to incorporate thermodynamical information into the classification while taking advantage of rapidly updating radar data. We tested three widely used supervised ML (DT, RF, and SVM) algorithms. The precipitation type (CLR, RA, MIX, and SN) was considered as a response. To select the important predictors, we conducted a variable importance analysis using a total of 9 candidate predictors from polarimetric radar and thermodynamical field. Subsequently, the selected 7 predictors were used in the ML models. We optimized the ML models by tuning hyperparameters and evaluated the accuracy of the three optimized models. RF was chosen as the best model.

As expected, the variable importance analysis confirmed that $T_w$ and $Thick_{1000-850}$ are essential factors for distinguishing between the precipitation type (RA, MIX, and SN). This study also found the higher importance of HGT compared to other polarimetric variables. This was because HGT acted as a useful factor in discriminating no precipitation (CLR) from the precipitation type (RA, MIX, and SN).

Our fine-tuned ML models were successful in discriminating CLR, RA, MIX, and SN. The RF showed the best performance with the highest POD and the lowest FAR for all precipitation types. In particular, the RF showed the highest CSI for CLR (0.784), RA (0.846), MIX (0.283), and SN (0.840). In addition, we compared the proposed model with the optimized Matsuo scheme which is currently in operation. The results demonstrated that the ML algorithm outperformed the optimized Matsuo scheme. Despite the better performance compared to the optimized Matsuo scheme, the relatively lower score of MIX compared to other types suggests a classification of MIX remains a challenge (CSI of 0.283).

We further applied the RF to the operational S-band radar network to check the spatial distribution of the classification. The ground precipitation type was estimated at each radar grid point at a high spatial resolution (500 m). The results were generally reasonable as they showed not only the continuity in both time and space but also a good agreement with the ground truth. It should be mentioned that the ML technique is computationally efficient so the proposed model would be useful for solving realtime classification problems in other applications.

**Author Contributions:** Conceptualization, G.L. and K.S.; methodology, K.S., J.J.S. and G.L.; software, K.S. and J.J.S.; validation, K.S., J.J.S. and G.L.; formal analysis, K.S., J.J.S. and G.L.; investigation, K.S., K.K. and G.L.; resources, K.S. and K.K.; data curation, K.S. and K.K.; writing—original draft preparation, K.S.; writing—review and editing, K.K., J.J.S. and G.L.; visualization, K.S., K.K. and J.J.S.; supervision, G.L.; funding acquisition, G.L. All authors have read and agreed to the published version of the manuscript.

**Funding:** This work was supported by the National Research Foundation of Korea (NRF) grant funded by the Korea government (MSIT) (No. 2021R1A4A1032646). This research was supported by Basic Science Research Program through the National Research Foundation of Korea (NRF) funded by the Ministry of Education (NRF-2021R1A6A3A13042215). This work was funded by the Korea Meteorological Administration Research and Development Program under Grant KMI2022-00310.

**Data Availability Statement:** The data presented in this study are available on request from the corresponding author.

**Acknowledgments:** We thank the Weather Radar Center for maintaining the weather radar and for providing data. We also appreciate students and researchers in CARE, KNU for constructive discussions.

**Conflicts of Interest:** The authors declare no conflict of interest.

## Appendix A. The Optimized Matsuo Scheme

In this section, we presented how the optimized Matsuo scheme classifies the precipitation type. The procedure is described in [15], but we restate the methodology here since [15] is not written in English. There are two steps to determine the precipitation type: (1) discriminate snow and rain using $Thick_{1000-850}$ and proceed to step 2 for the undetermined types, and (2) determine the final precipitation type based on the criteria as a function of RH and $T_s$. The criteria for each step are summarized in Table A1. Additionally, see their comprehensive figure (Figure 7 in [15]) which overlays the step 2 criteria.

**Table A1.** Criteria of the optimized Matsuo scheme to determine the precipitation type. Adapted from [15] with permission from Korean Meteorological Society.

| Step | Criteria | Precipitation Type |
|:---:|:---:|:---:|
| 1 | $Thick_{1000-850} < 1281$ | SN |
| | $1281 \leq Thick_{1000-850} \leq 1297$ | (Proceed to step 2) |
| | $1297 < Thick_{1000-850}$ | RN |
| 2 | $RH \geq 75$ and $T_s \leq 0.9$ and $RH < (-100/13) \times T_s + 102.5$ | SN |
| | $T_s > 0.9$ and $RH < (-100/13) \times T_s + 89.5$ or $T_s \leq 0.9$ and $RH < 75$ | |
| | $RH \geq (-100/13) \times T_s + 89.5$ and $RH < (-100/13) \times T_s + 100$ and $RH < (-12) \times T_s + 120$ and $T_s > 0.9$ | MIX |
| | $T_s > 0.9$ and $RH \geq (-100/13) \times T_s + 100$ and $RH < (-12) \times T_s + 120$ or $T_s \leq 0.9$ and $RH \geq (-100/13) \times T_s + 102.5$ | |
| | $RH \geq (-12) \times T_s + 120$ | RN |

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
