# Peer review of "Classification of Precipitation Types Based on Machine Learning Using Dual-Polarization Radar Measurements and Thermodynamic Fields"

_remotesensing, doi:10.3390/rs14153820_

Round 1
Reviewer 1 Report
The paper compares three machine learning models (decision trees, random forest and support vector machine) for the classification of precipitation particles (rain, snow, mix phase) using dual-polarisation radar measurements (Zh, rhohv, ZDR and KDP) and meteorological data from numerical weather prediction models (Tw, Thick, Ts, gamma_low). The paper is novel and well-written. There are however a few comments (see below) that the authors need to consider before the paper is accepted for publication.
L38. “the thickness of the layer … “ which layer?
Introduction. The authors mentioned several papers that used different approaches (fuzzy logic, Bayesian, etc) for the classification of precipitation particles using polarimetric radar measurements. It would be worth to mention some of the early papers in this field related to the microphysics of precipitation types and classification such as Vivekanandan et al (1999) and Straka et al, (2000). Some of these methods for the classification of precipitation types were demonstrated using fuzzy logic classifiers (Liu et al, 2000; Rico-Ramirez et al, 2005). The authors may also want to cite the paper on verification of precipitation types by Pickering et al (2021) who used disdrometer observations to validate radar-based surface precipitation types.
Line 104. the authors use precipitation types obtained from trained observers at KMA. I think it is important to discuss how the observers come up with each precipitation type (e.g. do they follow some specific rules or is it just eye observation?); what’s the temporal resolution of these observations? how do you deal for instance when there is maybe some particular precipitation type for a few minutes followed by another one a few minutes later. In addition, given the fact that there are 23 stations in different parts of south korea and that different trained observers are doing the classifications, how do you know that the quality of the precipitation types is consistent across all the stations?
Line 114. Need to be more explicit of what MIX includes. Is graupel and sleet part of the MIX classification? What is ‘No precipitation’ (are these echoes due to non-meteorological echoes e.g. ground clutter?)? Need to be more specific.
Line 147. Is T the same as Ts?
Line 157. Maybe worth to include the equations used to compute Thick1000-850 and gamma_low.
L239. I think you should optimise both, mtry and ntree using a 2D OOB plot rather than assuming a fixed value for ntree.
Fig 10. what are the units of the y axes?
The method(s) assume that outputs of NWP models are available. Perhaps the authors should comment on the applicability of these methods if NWP data are not available.
References
Straka, J.M., Zrnić, D.S. and Ryzhkov, A.V., 2000. Bulk hydrometeor classification and quantification using polarimetric radar data: Synthesis of relations. Journal of Applied Meteorology, 39(8), pp.1341-1372.
Vivekanandan, J., Zrnic, D.S., Ellis, S.M., Oye, R., Ryzhkov, A.V. and Straka, J., 1999. Cloud microphysics retrieval using S-band dual-polarization radar measurements. Bulletin of the american meteorological society, 80(3), pp.381-388.
Liu, H. and Chandrasekar, V., 2000. Classification of hydrometeors based on polarimetric radar measurements: Development of fuzzy logic and neuro-fuzzy systems, and in situ verification. Journal of Atmospheric and Oceanic Technology, 17(2), pp.140-164.
Rico‐Ramirez, M.A., Cluckie, I.D. and Han, D., 2005. Correction of the bright band using dual‐polarisation radar. Atmospheric Science Letters, 6(1), pp.40-46.
Pickering, B.S., Best, S., Dufton, D., Lukach, M., Lyth, D. and Neely III, R.R., 2021. Improving Observations of Precipitation Type at the Surface: A 5-Year Verification of a Radar-Derived Product from the United Kingdom’s Met Office. Journal of Hydrometeorology, 22(6), pp.1487-1505.
Reviewer 2 Report
My comments as following:
1. Explain the importance of classification of precipitation into RA, SN, MIX and CLR?
2. Can we apply the same diagram in the case of using a unipolar radar?
3. We suggest that the authors add a scheme to improve explaining your technique better.
4. Use a larger database for the application.
5. The authors need to present the optimized Matsuo scheme better. Indeed, this will make it possible to better understand its contribution in the evaluation of the method adopted.
6. The authors must improve the discussion and conclusion. Indeed, the authors insist on the important parameters of this classification (T and Thick1000-850) without indicating the importance of this classification or the continuation to be given to it.
Reviewer 3 Report
Dear Authors
First, I would like to congratulate you on the great work you have done. Increasing knowledge in the meteorological area using data from sophisticated equipment (weather radars) and applying artificial intelligence methods will always be a challenge.
I have just a couple of observations/questions.
1. At what altitude are the radar network located? and the ground-based rain gauges? I ask this because of the vertical advection effect. That is, the effect of wind on the meteors (rain, snow, etc.) because the precipitation observed by the radar does not fall vertically to the ground.
Please explain this process/effect and how you dealt with it in the study. If you did not consider it, this analysis may help improve the predictions for the next studies.
2. Improve figures 6 and 7. The font size of the titles, captions, and legends is not in accordance with the figures and manuscript.
Similar case for figures 8 and 9. The identification (letters a), b)...) does not have a good scale.
Regards.
